# Staying Ahead in the MOOC-Era by Teaching Innovative AI Courses

**Patrick Glauner** [1]

## Abstract

As a result of the rapidly advancing digital transformation of teaching, universities have started to face major competition from Massive Open Online Courses (MOOCs). Universities thus have to set themselves apart from MOOCs in order to justify the added value of three to five-year degree programs to prospective students. In this paper, we show how we address this challenge at Deggendorf Institute of Technology in ML and AI. We first share our best practices and present two concrete courses including their unique selling propositions: Computer Vision and Innovation Management for AI. We then demonstrate how these courses contribute to Deggendorf Institute of Technology's ability to differentiate itself from MOOCs (and other universities).

## 1. Introduction

Over the past decade, access to knowledge has fundamentally changed. This process began around 2011, when Stanford professors Andrew Ng, Sebastian Thrun and others made their AI courses available to everyone through online courses (Ng & Widom, 2014). This type of course is often referred to as a Massive Open Online Course (MOOC). Popular MOOC platforms include Coursera, Udacity, edX, Udemy and others. Until 2011, AI could generally only be studied in a limited number of available university courses or from books or papers. Furthermore, those resources were mainly available in developed countries. As a consequence, potential learners in emerging markets could not easily access respective resources. Due to MOOCs, the so-called "democratization of AI knowledge" has begun to fundamentally change the way we learn and has given rise to new AI superpowers, such as China (Lee, 2018).

We argue in Section 2 that MOOCs have now given many universities and professors serious competitors. It can also be assumed that this competition will intensify even further in the coming years and decades. In order to justify the added value of three to five-year degree programs to prospective students, universities must differentiate themselves from MOOCs in some way or other.

In this paper, we show how we address this challenge at Deggendorf Institute of Technology (DIT) in our AI courses. DIT is located in rural Bavaria, Germany and has a diverse student body of different educational and cultural backgrounds. Concretely, we present two innovative courses (that focus on ML and include slightly broader AI topics when needed):

- Computer Vision (Section 3)
- Innovation Management for AI (Section 4)

In addition to teaching theory, we put emphasis on real-world problems, hands-on projects and taking advantage of hardware. Particularly the latter is usually not directly possible in MOOCs. In this paper, we share our best practices. We also show how our courses contribute to DIT's ability to differentiate itself from MOOCs and other universities.

## 2. MOOCs Have Become Game Changers

In addition to courses on AI, a variety of other courses on almost any topic have emerged on various MOOC platforms over the past decade. Those courses enable learners to study high-quality content from renowned professors, remotely, at their own speed and at little or no cost. Furthermore, collaborations with renowned universities and industry partners have emerged. Some MOOC platforms offer career coaching, too. Companies have also launched collaborative programs with MOOC platforms to train their employees.

There are plenty of examples of professionals who have found new, high-paying jobs in various industries within a short period of time after completing hands-on MOOCs, for example in the news (Lohr, 2020) or on LinkedIn. This is particularly true for IT, a sector that has traditionally been open to lateral entrants and autodidacts. In recent years, MOOCs have therefore become steadily more established. This trend has also been further consolidated during the COVID-19 pandemic, as millions of people around the globe have been undergoing retraining (Bylieva et al., 2020).

[1] Department of Applied Computer Science, Deggendorf Institute of Technology, Deggendorf, Germany. Correspondence to: Patrick Glauner <patrick.glauner@th-deg.de>.

*Proceedings of the 2ⁿᵈ Teaching in Machine Learning Workshop*, PMLR, 2021. Copyright 2021 by the author(s).

In summary, universities will be facing the following challenges in the coming years:

1. In just a few years many very good high school graduates could decide against the traditional completion of a university degree program (if they do not aim for an academic career). They would then rather acquire all necessary practical skills through MOOCs within a few months or perhaps a year. In parallel, they could also gain practical experience by working part-time or founding startups. As a consequence, they could quickly get excellent jobs and outperform traditional university graduates on the jobs market.

2. Due to the demographic change (Magnus, 2012) and the potential lack of qualified new students, many universities in developed countries may become unable to maintain their current size. In view of the return on investment, politicians or administrators may thus probably sooner or later start thinking about closing individual departments or even entire universities.

3. Many (non-computer science) degree programs have so far only taught traditional content, with little or no link to the digital transformation and automation through AI. If this important content continues to go unnoticed in education, those degree programs will almost certainly train their students for unemployment.

Universities must face up to these challenges, which also provide many opportunities, though. By addressing these challenges and also taking even more advantage of their assets, such as enabling students to collaborate physically and using on-site facilities, universities could emerge even stronger from that competition. Most importantly, universities must differentiate themselves from MOOCs. In the following sections, we show how we address these challenges by teaching cutting-edge real-world content and taking advantage of physical university infrastructure. We also actively promote our courses through social media, press releases and other channels in order to attract more prospective students. In addition, our courses are open to students of other departments, including electrical engineering, mechanical engineering, healthcare or business. This allows us to support them in learning the tools of the 21st century that they need in order to actively contribute to the digital transformation of their disciplines.

## 3. Computer Vision Course

Popular MOOC platforms offer a number of excellent courses[1] on computer vision (CV). In order to survive in

---

[1]These include, but are not limited to, the following courses: http://www.udacity.com/course/computer-vision-nanodegree--nd891, http://www.coursera.org/learn/computer-vision-basics.

international competition, the content of a today's university CV course must meaningfully differentiate itself from those by offering unique selling propositions. Based on these principles, we have started to teach this novel course in 2020 at DIT. Note that there is a separate deep learning course taught by a different professor in our department. Most students take both courses in parallel and have previously taken an introductory machine learning course.

### 3.1. Content

We provide students with a broad and deep background in CV. That is why we discuss both, traditional and modern neural network-based CV methods. In practice, successful CV applications tend to combine both approaches (O'Mahony et al., 2019), in particular when only a limited number of training examples are available (Ahmed & Islam, 2020). Concretely, we discuss the following topics in the first half of the term:

- Introduction: applications, computational models for vision, perception and prior knowledge, levels of vision, how humans see
- Pixels and filters: digital cameras, image representations, noise, filters, edge detection
- Regions of images: segmentation, perceptual grouping, Gestalt theory, segmentation approaches, image compression by learning clusters
- Feature detection: RANSAC, Hough transform, Harris corner detector
- Object recognition: challenges, template matching, histograms, machine learning
- Convolutional neural networks: neural networks, loss functions and optimization, backpropagation, convolutions and pooling, hyperparameters, AutoML, efficient training, selected architectures
- Image sequence processing: motion, tracking image sequences, temporal models, Kalman filter, correspondence problem, optical flow, recurrent neural networks
- Foundations of mobile robotics: robot motion, sensors, probabilistic robotics, particle filters, SLAM
- Advanced topics: 3D vision, generative adversarial networks, self-supervised learning

In the second half of the term, students work in groups of 1 to 4 members on a CV project.

### 3.2. Unique Selling Propositions

This course differentiates itself from other CV courses, in particular MOOCs, as follows:

1. Most CV courses taught on MOOC platforms or at universities only include smaller, isolated problems that can be implemented on almost any commercially available computer or by using cloud services. This

course includes a larger real-world project in the second half of the term instead. Students choose a CV project of their choice, in which they also apply agile project management and use respective tools. In order to provide students with a real added value of a physical university course, they are highly encouraged to use the NVIDIA Jetbot platform depicted in Figure 1.

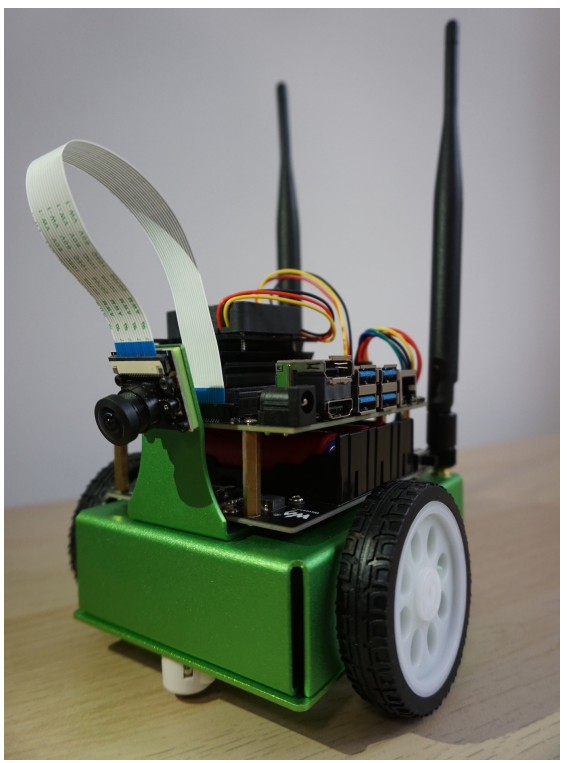

Figure 1. The NVIDIA Jetbot mobile robot platform used in the projects. Find more information at http://www.github.com/NVIDIA-AI-IOT/jetbot.

It possesses a camera and efficiently executes CV algorithms on its NVIDIA Jetson GPU. By using this platform, students can not only better understand the course content. Rather it enables them to experience how these algorithms behave in the real world. During the COVID-19 pandemic, they could take the robot kits home in order to work on their projects remotely.

2. We cover challenging content that is more complex than in most available MOOCs: We first reviewed CV courses at introductory and advanced levels of international top universities, including Stanford, MIT and Imperial College London. We then selected the topics that we find most relevant to solving real-world problems. Furthermore, we present these topics in a more understandable way and include additional revisions of the underlying concepts. Like this, we also make this

course more accessible to students of other disciplines.

### 3.3. Outcomes and Students' Feedback

19 students of different degree programs signed up for the first iteration of this course. In total, they implemented 10 projects in groups of 1 to 3 members.

About half of the projects used a NVIDIA Jetbot. Those projects included object following and simultaneous localization and mapping (SLAM). The other projects included a face mask detector and a clothes classifier. We find the coin counter depicted in Figure 2 particularly worth mentioning though.

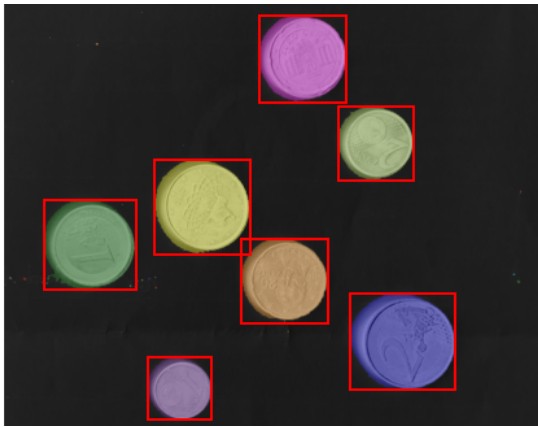

Figure 2. Coin counter. Image courtesy: Patrick Gawron and Achot Terterian.

It first applies object segmentation and detection to a photo that contains an arbitrary number of coins. It then aggregates the amounts of the individual coins. The underlying ML model also handles multiple currencies in the same photo. In the project presentation, the group also discussed how they solved the challenge of collecting a data set of coins that includes a variety of angles, conditions, reflections and currencies.

We received quantitative and qualitative feedback from students through a formal course evaluation. The overall feedback was a 1.3 on a scale 1 to 5 where 1 is the best. However, a few students suggested a longer introduction to deep learning frameworks for the first half of this course. They would then have been able to start working on their projects quicker in the second half. In the second iteration, we have therefore added an extended introduction to deep learning frameworks.

## 4. Innovation Management for AI Course

In recent years, many companies have started to invest in ML and AI to stay competitive. However, the sad truth is that some 80% of all AI projects fail or do not result in any financial value (Nimdzi Insights, 2019). That is a serious concern because there is clearly an acute need in industry for experts who have a comprehensive knowledge of what needs to be done so that AI adds value to businesses. In our view, one of the underlying causes is the way AI is taught in universities, as most courses cover only purely methodological and engineering aspects of AI. We are convinced that professors need to address this problem by also enabling students to think in a broader and business-oriented sense of AI innovation management. At DIT, we therefore started to teach this novel and internationally unique course in 2020.

### 4.1. Content

We discuss a range of challenges, both technical and managerial, that companies typically face when using AI (Glauner, 2020). We first look back at some of the historic promises, successes and failures of AI. We then contrast them to some of the advances of the deep learning era and contemporary challenges. Concretely, we discuss the following topics:

- Introduction: how AI is changing our society, selected examples of successful and unsuccessful AI projects and transformations
- History and promises of AI: Dartmouth conference, AI from 1955 to 2011, AI winters
- Deep learning era: breakthroughs, DeepMind, promises and hypes, no free lunch theorem, AI innovation in China, technological singularity
- Contemporary challenges: regulation, explainable AI, ethics
- AI transformation of companies: opportunities, challenges, best practices, roles, data strategy, data governance

We offer this course as an intensive course. On day one, we teach the content above. On the following two days, students work on a case study on how to successfully implement AI in a company of their choice. They present the outcomes of their case study on day four.

### 4.2. Unique Selling Propositions

This course differentiates itself from other courses, in particular MOOCs, as follows:

1. During an intensive online search for related courses, we only found introductions to AI for managers[2]. How-

ever, we did not find any business-related courses for AI experts. In this course, we bridge that gap.
2. Students learn respective best practices along the entire AI value chain and how these lead to productively deployed applications that add real value. They work on a case study on how specific AI use cases are implemented in companies, what challenges may be encountered and how they may be solved.

### 4.3. Outcomes and Students' Feedback

21 students of different degree programs signed up for the first iteration of this course. In total, they worked on 11 case studies in groups of 2 to 4 members.

Most of the students who took this course are computer scientists studying in a part-time continuing education AI degree program. We received very positive feedback from them as they could include in their case studies some of the current challenges they face at work. A few business students also took this course as they were eager to learn more about AI. They contributed their in-depth business knowledge to the case study presentations. This turned out to be a valuable experience for the computer scientists.

We could, however, not quantitatively assess this course yet. Our university's course evaluation scheme does not include intensive courses. We are planning to address this issue in the future with an unofficial course evaluation.

## 5. Conclusions

Universities are facing major challenges as a result of the rapidly advancing digital transformation of teaching. These include in particular competition from Massive Open Online Courses (MOOCs). This transformation is further being accelerated by the demographic change in developed countries and could result in a dwindling number of potential students in the near future. However, if universities address those challenges swiftly, ambitiously and sustainably, they can even emerge stronger from this situation by providing better and modern courses to their students. In this paper, we showed how we address those challenges in AI education at Deggendorf Institute of Technology. Concretely, we teach innovative and unique courses on computer vision and innovation management for AI. We shared our best practices and how our courses contribute to Deggendorf Institute of Technology's ability to differentiate itself from MOOCs and other universities.

Both courses are currently being offered again. The number of students that signed up has more than doubled. Our courses are thus positively perceived by students.

---

[2]These include, but are not limited to, the following courses: http://www.udacity.com/course/ ai-for-business-leaders--nd054, http://www. udemy.com/course/intro-ai-for-managers/.

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
