# OpenReview forum: "Staying Ahead in the MOOC-Era by Teaching Innovative AI Courses"
_ecmlpkdd.org/ECMLPKDD/2021/Workshop/TeachML — TeachML 2021_

### Official Review · Reviewer_GSyo · 2021-07-14
**Interesting challenges due to the rise of MOOCs are not followed up by a thorough discussion**

**Rating:** 6
**Confidence:** 4

**Review:**

The paper describes challenges for universities that arise with the growing quality and quantity of MOOCs and presents two own courses, one on computer vision, one on Innovation Management for AI.

Overall, the paper is written in a comprehensive style and structured clearly. Chapter 2 raises very interesting thoughts on the challenges for universities stemming from the growing amount of MOOCs. Unfortunately, these challenges are not further and thoroughly discussed, but instead, the following chapters describe two own courses which for me seems to be more an advertisement of those courses than a real contribution to the scientific discussion on the challenges of MOOCs. In my opinion, the paper would be a lot stronger, if the stated challenges in Chapter 2 would have been followed up by a thorough discussion of how universities may tackle those challenges and which potential benefits for university education the rise of MOOCs may have. However, since I think the discussion on MOOCs at the Teaching ML Workshop could be very interesting, I recommend accepting the paper - regardless of the above objections.

Pros:
- The challenges due to MOOCs are a very relevant topic in my opinion
- Clear structure and clean language

Cons:
- Discussion of those challenges is too short
- Relevance of the two described courses is not clear to me
- Number of students in the described courses is very small, hence the evaluation and feedback are not very significant

Some minor comments and further thoughts:
- The last sentence of the abstract points at a demonstration "how these courses contribute to Anonymous University's ability to differentiate itself from MOOCs (and other universities)": I could not find this section; after describing the courses, the paper ends with a conclusion.
- Section 2:
	- I am not sure if it is true that potential students would decide against a university career and educate themselves just via MOOCs: At least in some countries, a university degree is still important to get a (good) job in the industry - and definitely necessary for an academic career.
	- How should students know which MOOCs they would have to take if there are so many available? A university curriculum defines that.
	- Let's say it would be true that universities would lose students who would rather educate themselves via MOOCs for an industry career: Couldn't that be a chance for universities to get back to their roots and educate students more for an academic career and wouldn't this be a good development?
- In universities, students can directly interact with teachers, get individual feedback, and so on. MOOCs (until now) can't do this. This is a unique selling point of university education - which perhaps should be exploited better in some parts of university education.
- Chapter 4: "We are convinced that professors need to address this problem by also enabling students to think in a broader and business-oriented sense of AI innovation management." I do not agree and do not think that business orientation is the core responsibility of university education.

---

### Official Review · Reviewer_8ALG · 2021-07-16
**Interesting angle on course design, but might be missing a huge advantage**

**Rating:** 6
**Confidence:** 4

**Review:**

This submission describes 2 university courses developed and taught by the authors -- a semester-long course on Computer Vision, and 4-day "intensive course" on Innovation Management for AI. The course are described in the context of competing against ML and AI MOOC courses offered by various platforms.

The authors did a good job of describing the courses, and the content advantages they hold over MOOC courses. However, I thought that the authors missed out on (or at least glossed over) a huge advantage that their courses provide over MOOCs -- real-world collaboration with other students on a project. The CV course (in Section 3.2) specifically mentions students applying agile project management. For the Innovation Management course, Days 2 and 3 are devoted to group case study work and then Day 4 is a presentation of their work. This type of group work is a major advantage, but the comparison to MOOCs appears limited to content.

I also thought that the language often used was a bit over-the-top. For example, in Lines 071-076 (Column 1), the authors state that non-CS degree programs that do not incorporate digital literacy and AI will "almost certainly train their students for unemployment".

All of that being said, I think that hearing about the courses and the MOOC-influenced strategy involved in their creation would be an interesting addition to the workshop.

---

### Decision · Program_Chairs · 2021-07-21

**Decision:**

Accept

**Comment:**

Congratulations! The reviewers agree that this paper should be accepted.

Camera-ready version is due August 18, 2021. As you prepare the camera ready version, please take the reviewers comments into consideration.

We look forward to your participation at the workshop on September 13, 2021. We invite you also to join us for the satellite event on September 08, 2021. Schedules for both the workshop and the satellite event will be forthcoming.